

# Loss assessment of building and content damages from potential earthquake risk in Seoul, Korea

Wooil Choi[1], Jae-Woo Park[2] and Jinhwan Kim[2,3]

[1]Research Institute, LIG System, 442, Bongeunsa-ro, Gangnam-gu, Seoul, 06153, Korea

[2]Department of Civil and Environmental Engineering, Hanyang University, 222 Wangsimni-ro, Seongdong-gu, Seoul, 04763, Korea

[3]Multi Disaster Countermeasures Organization, Korea Institute of Civil Engineering and Building Technology, 283, Goyang-daero, Ilsanseo-gu, Goyang-si, Gyeonggi-do, 10223, Korea

*Correspondence to*: Jinhwan Kim (goethite@kict.re.kr)

**Abstract.** After the 2016 Gyeongju earthquake and the 2017 Pohang earthquake struck the Korean peninsula, securing financial stability for earthquake risk has become an important issue in Korea. Many domestic researchers are currently studying potential earthquake risk. However, empirical analysis and statistical approach are ambiguous in the case of Korea because no major earthquake has ever occurred on the Korean peninsula since Korean Meteorological Agency started

monitoring earthquakes in 1978. This study focuses on evaluating possible losses due to earthquake risk in Seoul, the capital of Korea, by using catastrophe model methodology integrated with GIS (Geographic Information System). The building information such as structure and location is taken from the building registration database and the replacement cost for building is obtained from insurance information. As the seismic design code in KBC (Korea Building Code) is similar to the seismic design code of UBC (Uniform Building Code), the damage functions provided by HAZUS-MH are used to assess the damage

state of each building in event of an earthquake. 12 earthquake scenarios are evaluated considering the distribution and characteristics of active fault zones in the Korean peninsula, and damages with loss amounts are calculated for each of the scenarios.

## 1 Introduction

On November 15, 2017, an earthquake of M 5.4 on Richter scale hit the northern region near Pohang city located in

southeastern part of the Korean peninsula. After 5.8 Gyeongju earthquake in 2016, it was the second strongest recorded earthquake in Korea since the monitoring began in 1978 (Fig. 1).

(Figure 1 is about here)

The earthquakes occurred in Gyeongju and Pohang are expected to be caused by Yangsan fault zone which is classified as active fault on the Korean Peninsula, which has the ability to generate a maximum of 7.0 M earthquake according to Kyung



(2010) and MPSS (2012). If earthquake of M 6.0, close to the Gyeongju and Pohang earthquakes, occurs in or near Seoul, where major industrial and commercial facilities are concentrated, huge loss that has never been experienced in the past might occur. Especially, disaster risk financing industry such as the insurance can subject to catastrophic damage. According to the Natural Disaster Reduction Project report prepared at the request of the MPSS (2015), 2.76 million people may lose their life

and 2,848 billion dollars of economic loss including indirect loss such as business interruption may occur if an earthquake of M 7.0 strikes Seoul. However, as this report relies on the HAZUS-MH for most of the analysis data such as replacement cost of property and seismic characteristics of earthquake, the estimated result may differ from actual damage loss amount in Korea. This study uses catastrophe model methodology to predict loss and damage of buildings and contents from the potential earthquake that can occur in Seoul. The detailed information of approximate 6.3 million buildings across Seoul is acquired

through the building registration database. The replacement cost of each building and contents are statistically estimated by using insurance database which is classified by occupancy to meet the reality of Korea.

## 2 Methodology

Predicting loss amount of potential disaster using catastrophe model differs from the actuarial approach model. While the actuarial technique estimates the loss based on empirical data, the catastrophe model generates disaster scenarios based on

scientific understanding of disasters and assesses the loss amount from event scenario. For possible quakes in Korea, it is appropriate to use the catastrophe model for predicting losses because empirical data from earthquakes on the Korean peninsula is too scarce to enable actuarial processing.

The definition and procedure of catastrophe model can marginally differ between researchers or suppliers but the conventional procedure can be illustrated as Fig. 2.

(Figure 2 is about here)

As shown in the above figure, the catastrophe model has a four step process. The first step is to build information database of property which may be exposed to disaster. However the exposure data sets in previous studies is typically available at

relatively coarse resolutions because it is accompanied by difficulties related to limited resources or privacy issues, among others (Dell'Acqua et al., 2012, Figueiredo and Martina, 2016). In order to overcome these limitations, this study used the building registration database of Korea to build exposure data set. The detail information of the building must be recorded, which is registered in the building registration database whenever the building is constructed or reconstructed according to the Building Act in Korea. In this study, the detail information needed to evaluate the vulnerability of all buildings in Seoul was

extracted from the building registration database. The extracted data are classified into 36 the structure types, 33 occupancies and divided into 3 seismic codes estimated based on comprehensive consideration of the construction year, total building area and occupancy.



The second step, called hazard module, generates a physical hazard map from a simulated event of disaster. For example, the peak ground acceleration can be represented as hazard intensity in an earthquake hazard map. The seismic events are usually generated by stochastic methodologies such as Monte Carlo simulation. However, it has not been less than 40 years since the earthquake began to be monitored on the Korean Peninsula, and there was no large-scale earthquake in Seoul during the monitoring period. In this study, synthetic earthquakes were generated considering the activity of the active faults passing through Seoul, the seismic hazard map was prepared by selecting the attenuation relation that most closely resembles the Gyeongju earthquake and the Pohang earthquake among a lot of attenuation relations proposed by many domestic and abroad researchers.

The third step is to prepare vulnerability module to assess damage state of individual properties by combining the information of exposed property and hazard intensity. The probabilities of each damage states of the building should be estimated from spectral displacement of building due to seismic impact in the vulnerability module. The spectral displacement is determined by performance point, which is the intersection of the demand curve and the capacity spectrum.

The final step, the financial module, is to quantify the damage of individual buildings into a monetary loss to predict a total loss amount. In order to estimate the repair cost of a building due to a seismic impact, it is necessary to ascertain the replacement value or the current asset of the building calculated in cost mode. In this study, the values of building, contents and inventories of representative building in each categories were estimated by statistical processing of appraisal data for insuring property.

### 3 Construction of exposure information

The detailed information of each building such as location, structure, size, floor area, construction year, occupancy, etc. affecting the seismic response is obtained from computerized database of building registration records. There are presently 6.3 million buildings within Seoul city. These buildings are classified as residential (76% of the total number of buildings), commercial (20.3%), industrial (0.5%) and others (3.2%) that includes government and education institute buildings. Residential are predominantly dominated by masonry structure of less than five stories, and concrete structures are dominant in commercial. 82% of the buildings in Seoul were built before 1988 when seismic code of building began to be considered. Table 1 summarizes the statistical characteristics of buildings in Seoul.

(Table 1 is about here)

The replacement costs for each building and content are estimated based on statistical processing of 1,500 records of asset evaluation data for property insurances and construction cost table issued by Korea Appraisal Board. On processing, the total replacement cost of buildings and contents is estimated to be about 900 billion dollars in Seoul and approximately 72% of





buildings in Seoul were estimated to have replacement cost between USD 100,000 to 1,000,000. The indirect costs and losses attributed to land and intangible assets and business interruption are not considered in this study.

## 4 Hazard Assessment

### 4.1 Scenario selection

In the Circum-Pacific seismic zone, Korea has been safer and less prone to quakes as compared with its neighboring countries. However, many domestic researchers insist that there are two representative active faults in Korea. One of these, the Yangsan Fault, caused the Gyeongju earthquake. The second fault, Chugaryeong Fault is centrally located on the Korean peninsula (Choi et al., 2012; MPSS, 2012; Chung et al., 2014). Chugaryeong Fault crosses the eastern side of Seoul and is believed to have caused 2010 earthquake, of M 3.0, in Seoul.

(Figure 3 is about here)

The Chugaryeong Fault has similar activity to Yangsan Fault which has the capacity to cause an earthquake of M 7.0, also most earthquakes in Korea occur or are likely to occur at focal depth of about 10 km (Lee, 2010; MPSS, 2012). Based on this, earthquakes of M 4.0 to 7.0 occurring at focal depths of 10 to 20 km at the southeast of Seoul due to activity of the Chugaryeong Fault are selected as event scenarios of this study.

### 4.2 Construction of hazard map and response spectrum

To construct each hazard map from each earthquake event scenario, it is important to understand the attenuation relationship of ground motions from epi-central distance. The ground motion can be characterized by PGA and spectral response based on a response spectrum shape.

A lot of experimental attenuation formulas for estimating PGA have been developed by means of regression analysis (Atkinson and Boore, 1997; Toro et al., 1997; Atkinson and Silva, 2000; Lee and Kim, 2002; MPSS; 2012). However, in choosing the attenuation formula, a careful approach is needed since the effect of the formula is very large on estimating amount of earthquake loss. MPSS proposed three attenuation formulas for Korean Peninsula, which are expressed with the equation as following Eq. (1), and the values of the coefficients are shown in table 2.

$$\ln S = C_0 + C_1 M + C_2 \ln R + C_3 R, \tag{1}$$



Where, *S* is Peak Ground Acceleration (PGA), M is magnitude of earthquake and R is epi-central distance.

(Table 2 is about here)

The influence of the seismic attenuation equation on the seismic hazard map is very large, but the reliability of the attenuation equations presented so far remains controversial. In this study, we tried to utilize the results of domestic studies reflecting the seismic characteristics in Korea, and the Formula III, which is considered to be the most conservative formula because the attenuation of seismic wave is the least of the formulas proposed by MPSS (2012), is chosen for building the earthquake hazard map from the event scenario. The hazard maps according to the each scenario are shown in Fig. 4.  PGA in Seoul ranges from

0.06g to 0.7g in these scenarios which earthquakes of M 4.0 to 7.0 occurred at focal depths of 10 to 20 km.

(Fig. 4 is about here)

The severity of vibratory response of building to earthquake impact depends on relationship between the characteristics of

ground motion described as response spectrum (which has a different shape according to ground conditions) and structural characteristics of building. But since the design response spectrum currently used in Korea is based on the high seismicity region like California, a lot of domestic researchers insist that the spectrum is different from characteristics of the earthquake on the Korean peninsula (Kim et al., 1998; Han, 2003; Hwang et al., 2015; Lee and Ju, 2017). In general, the ground condition of Korea including Seoul is characterized by shallow bedrock, and the earthquakes occurred in Korea have a characteristic that

the duration of strong motion is shorter than one in high seismicity region. In the case of the Gyeongju earthquake, the strong motion with short duration of 0.1~0.2 second was also observed. As shown in Fig. 5, shape of the standard response spectrum is described by four transition points. S is Peak Ground Acceleration (PGA) and αA is the amplification factor at the short-period. Heo et al.(2018) calculated the shapes and transition periods of the response spectrums through regression analysis of the accelerations and spectral displacements of the Gyeongju and the Pohang earthquakes and found that the standard response

spectrum which had been used previously for seismic design in Korea was overly conservative in long period part. The factor adapted the spectrum in this study are set as shown table 3 below after comparing the spectra of earthquakes in Gyeongju and the Pohang.

(Fig. 5 is about here)

(Table 3 is about here)



**5 Assessment of building vulnerability**

**5.1 The status of seismic design code in Korea**

In 1988 when an earthquake occurred in Mexico, seismic design code in Korea were first mandated for building with six and more stories or floor area of 100,000 ㎡ or more, and then it was gradually expanded to be adapted seismic design code for all

5 buildings with three and more stories or floor area of more than 500 ㎡ through the revision of KBC in 2015. Nevertheless, 93.2% of all buildings in Korea did not apply seismic code and have more vulnerable characteristics to earthquakes (SMG, 2012; Choi, 2016).

The seismic design codes in KBC were established based on the Uniform Building Code (UBC), the Applied Technology Council (ATC) and the International Building Code (IBC) (SMG, 2012; Lee, 2015). While the seismic zone in UBC is divided

into six zones which have each regional factor defined as design peak ground acceleration (PGA), the zone in KBC is divided into two zones; zone I which includes Seoul area and zone II. The regional factor of zone I was 0.11 before 2009 but strengthened to 0.22 after that, and the seismic design code of buildings built in Seoul before 2009 is similar to zone 2A of UBC and the code of buildings built after 2009 is similar to zone 2B of UBC.

(Table 4 is about here)

**5.2. Application of damage function**

As mentioned above, since seismic design code of Korea is similar to the UBC and the ATC code, the damage functions proposed by HAZUS-MH can be applied to estimate building damage due to seismic impact. The damage function for each

20 building type in HAZUS-MH includes two types of damage curves; capacity curve and fragility curve. The capacity curve is used to determine peak building response from the capacity spectrum method. This method is a schematic procedure for comparing the capacity curve obtained by push-over analysis with the demand spectrum of ground motion on the Acceleration Displacement Response Spectrum (ADRS). Thus response spectrum has to be converted to demand spectrum for representing the relationship between spectral displacement and spectral acceleration. Eq. (2) proposed by HAZUS-MH can relate spectral

acceleration with spectral displacement for given period value (FEMA, 2013).

$$S_d = 9.8 \cdot S_a \cdot T^2, \tag{2}$$

Where, $S_d$ is spectral displacement (inches), $S_a$ is spectral acceleration (g) at a period (T, second)

The intersection of the capacity curve and the demand spectrum is a performance point which can evaluate the associated damage state for the structure and compare that damage state for different earthquakes (Fig. 6).





(Fig. 6 is about here)

The fragility curves estimate the probability of exceeding different damage states given peak building response represented as
5   spectral displacement or spectral acceleration at performance point. The damage state is divided into four states; Slight, Moderate, Extensive, and Complete. Each fragility curve is expressed as lognormal function defined by a median value of peak building response, corresponding to the mean threshold of associated damage state by a logarithmic standard deviation (β). The fragility curve for structural component of building uses spectral displacement (Sd) as peak building response and define the function of Eq. (3) and Fig. 7.

$$P[ds|S_d] = \Phi[\frac{1}{\beta_{ds}}\ln(\frac{S_d}{\bar{S}_{d,ds}})], \qquad\qquad (3)$$

Where, $\Phi(\cdot)$ is the standard normal distribution function and $\bar{S}_{d,ds}$ is median value of spectral displacement at which the building reaches the threshold of damage state.

(Fig. 7 is about here)

The non-structural components of the building are divided into drift-sensitive components and acceleration-sensitive components. In general, while architectural components such as interior or exterior wall are more drift-sensitive, the
mechanical and electrical components of building are acceleration-sensitive. Therefore the functions of inter-story drift is used to estimate damage state of drift-sensitive components and function of floor acceleration is used to estimate the damage state of acceleration sensitive components or contents in the building.

The capacity curve and the fragility curve in the HAZUS-MH are classified into high-code, moderate-code, low-code and pre-code buildings as per seismic design codes (FEMA, 2013). When all buildings in Seoul are classified comparing seismic
design code of the HAZUS-MH, it is estimated to approximately 91.7% pre-code, 5.4% low-code and 2.9% moderate-code buildings.

### 5.3. Calculation of loss ratio

Using estimates, which include the structural and nonstructural repair costs caused by building damage and the associated loss
of building contents and business inventory, provided by HAZUS-MH, the probability of exceeding different damage state for the each component can convert to loss ratio of replacement cost for evaluation of direct economic loss. Table 5 and 6 summarize the estimated mean loss ratio of building which includes structural and non-structural components, and contents





depending on occupancy and structure type. It is a common pattern that the majority of damage occur to low-rise residential building made of masonry in the Korean Peninsula where the earthquake characterized by strong short-period component is dominant. However this common pattern isn't clearly shown in the result of this study, and there are two main reasons for this. The first reason can be found that although the number of low rise residential building made of masonry is much higher, the

5    total asset value is much lower than the high rise residential building made of reinforced concrete like apartment, which is generally classified as luxury residence, while low-rise masonry house is classified as low priced residence in Seoul. The second reason is that the non-structural elements such as mechanical and electrical components, which are more vulnerable to ground shaking than structural components, in the buildings made of concrete and steel structure have higher proportion than in masonry.

(Table 5 is about here)

(Table 6 is about here)

Fig. 8 is a map that shows the loss ratio of each building in the Gangnam district, located 3km away from epi-center with M

15    4.0~7.0 and focal depth 10km. According to the results, if an earthquake of M 4.0 strikes southeast part of Seoul, damage to the residential buildings of pre-code start to occur and an earthquake of M 5.0 can damage almost all buildings due to ground shaking. And if an earthquake of M 6.0 occurs, office buildings of low-code begin to be damaged by seismic impact, and an earthquake of M 7.0 is estimated to cause an average 14.8% and 14.9% of total replacement cost of all buildings and contents respectively in Seoul.

(Fig. 8 is about here)

## 6. Estimation of loss amount

The total loss amount for each scenario can be simulated by replacement cost, seismic intensity, damage function and other

25    factors mentioned above. But since systematic data issues or biases across a portfolio can result in losses being consistently under- or over-simulated (LMA, 2017), the results need to be corrected by comparing empirical data. Linear Scaling Method (LSM), which is one of the common method to correct systematic errors, can be used to calibrate pre-simulated loss amount. LSM reflects the difference between pre-simulated results and observed results in the simulated results as shown by Eq. (4).

30    $$L_{cor,i,j} = L_{sim,i,j} + \left( \sum_{i=1}^{n} L_{obs,i,j} - \sum_{i=1}^{n} L_{sim,i,j} \right), \tag{4}$$



Where *i* is M of earthquake, *j* is focal depth. *Lcor,i,j is* the corrected loss, *Lobs,i,j* is the observed loss, and *Lsim,i,j* is the pre-simulated loss. There has never been an earthquake of M 4.0 or more in or near Seoul since earthquake monitoring began in 1978. Therefore, all pre-simulated results were inevitably corrected using empirical data of the earthquake of M 3.0 that

occurred near Seoul in 2010. The calibrated loss amounts for each scenario are summarized in Table 7. The total loss in the case of an earthquake of M 4.0 is estimated at 2.2 billion dollars. However, if the M of the earthquake increase to 7, the total loss is estimated to increase 58 times of M 4.0, reaching 126.6 billion dollars which is close to % 15 of total replacement cost for all buildings in Seoul.

(Table 7. is about here)

### 7. Conclusion

The existence of active fault zones on the Korean peninsula and recent quakes that affected Gyeongju and Pohang cities have made experts question whether current overall practices would still be adequate if a similar quake occurs in Seoul. And the concentration of major industrial and commercial facilities carries a significant inherent risk to cause catastrophic loss of life

and economy and significant administrative challenge for disaster management in Korea. The disaster management is divided into four phases; 1) mitigation, 2) preparedness, 3) response and 4) recovery. At each phase which has particular needs and problems, different strategy and support are required to force social resilience against each natural disaster. And it is also important that the activities at each phase generate virtuous cycle and assist in making each other to be stronger.

The development of insurance industry can be a good example to explain virtuous cycle in disaster management. The insurance

industry as disaster risk financing commonly plays a major role to secure financial stability for smooth recovery from natural disaster. However, it also helps these activities to perform more effective during the other phases such as mitigation, preparedness and response. The Sichuan earthquake of 2008 is in stark contrast with New Zealand earthquake of 2010 in terms of disaster management efficiency due to limited insurance penetration. The Sichuan earthquake of M 8.0 that occurred in China, where insurance penetration is relatively low, caused approximately 70,000 deaths, more than 370,000 injuries, and

127 billion dollars of economic loss. However, the insured loss was under 3% of the economic loss. On the other hand, the earthquake of M 7.1 that occurred in New Zealand, where insurance penetration is very high, caused only 2 injuries and 2.7 billion dollars of economic loss, which is more than 50% of economic loss that was covered from various insurance programs such as direct insurance, reinsurance and international financing market (WEF, 2011).

Most domestic insurers believe that it is impossible to predict loss amount from potential earthquake and it is difficult to

quantify the earthquake risk in Korea. This belief of insurers is a major obstacle to development of the earthquake insurance programs. However, as mentioned above, various studies required for the catastrophe model methodology have been either completed or in progress by various domestic researchers and a lot of database related to potential earthquake risk in Korea is

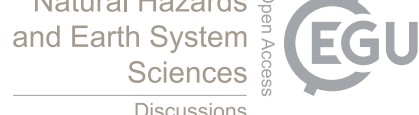



being accumulated. Compared to other studies, this study is differ in that real insurance information and building registration database are used to predict loss amount from potential earthquake. It not only helps advance the prediction process but also serves insurer to better understand and estimate the earthquake risk. This study shows that risk due to potential quakes in Korea is significant and insurance industry can support more detailed studies for better understanding of insurance risk and expanding

scope of current insurance practices for earthquake risks. Because of this, Insurance companies have an opportunity to further explore currently under tapped areas of business in property insurance.

**Acknowledgments**

This work was supported by the National Research Council of Science & Technology (NST) grant by the Korea government (MSIP) (No. CRC-16-02-KICT)

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





**TABLE**

Table 1: A summary of the statistical characteristics of buildings in Seoul.

| Classification | Distribution (%) |
|---|---|
| **Occupancy** | |
| Residential | 76.0 |
| Commercial | 20.3 |
| Industrial | 0.5 |
| Others | 3.2 |
| **Structure** | |
| Masonry | 49.3 |
| Reinforced Concrete | 43.5 |
| Steel | 0.2 |
| Wood | 5.3 |
| Others | 1.7 |
| **Floor area (100m$^2$)** | |
| ~ 1 | 20.6 |
| 1 ~ 2 | 23.0 |
| 2 ~ 3 | 16.3 |
| 3 ~ 5 | 15.9 |
| 5 ~ 10 | 13.9 |
| 10 ~ 30 | 5.7 |
| 30 ~ | 4.7 |
| **Number of Floors** | |
| 1 | 20.3 |
| 2~5 | 72.1 |
| 6 ~ 10 | 4.2 |
| 11 ~ 20 | 2.3 |
| 21 ~ 30 | 0.5 |
| 31 ~ | 0.1 |



**Table 2: Coefficient Value of Each Formula.**

| Formula | Coefficient | | | | Standard Deviation |
|---------|--------|--------|--------|--------|--------|
| | $C_0$ | $C_1$ | $C_2$ | $C_3$ | |
| Formula Ⅰ | 0.4853 | 1.2 | -0.8416 | -0.0061 | 0.8036 |
| Formula Ⅱ | 0.5577 | 1.2 | -0.8587 | -0.0062 | 0.7629 |
| Formula Ⅲ | 5.0244 | 0.5442 | -1.0020 | 0.0 | 0.1000 |

Source: MPSS (2012)





**Table 3: Factors of standard response spectrum in this study.**

| | Amplification factor at short periods ($\alpha_A$) | Transition Period (sec) | | |
|---|---|---|---|---|
| | | $T_0$ | $T_S$ | $T_L$ |
| Gyeongju Earthquake | 2.85 | 0.054 | 0.22 | 1.5 |
| Pohang Earthquake | 3.15 | 0.07 | 0.195 | 4.475 |
| This study | 2.8 | 0.06 | 0.2 | 3 |



**Table 4: Alteration of seismic design code in Korea.**

| Classification | | Seismic design code | | | |
|---|---|---|---|---|---|
| | | 1988 ~ 2000 | 2000 ~ 2005 | 2005 ~ 2009 | 2009 ~ |
| Reference basis | | UBC85 ATC3-06 | UBC85 ATC3-06 | IBC2000 | IBC2000 |
| Regional factor | Zone I | Gwangju-si / Gangwon-do / Jeollabuk-do / Gochang-gun / Jeollanam-do / Uljin-gun / Jeju-do | All area except zone II | | |
| | | 0.12 | 0.11 | 0.11 | 0.22 |
| | Zone II | All area except zone I | North Gangwon-do / Jellanam-do/ Southwest/Jeju-do | | |
| | | 0.08 | 0.07 | 0.07 | 0.14 |
| Seismic design object | | Building (>6 stories), Total floor (>100,000㎡) Floor area (>10,000㎡ sales facility), Assemble Facility (>5,000㎡) General hospital (>1,000㎡), Power plant, Public service facility | | Building (>3 stories) Total floor (>1,000㎡) | |





**Table 5: Estimated mean loss ratio of building based on occupancy type.**

| M | Focal Depth (Km) | Residential | | Commercial | | Industrial | | Others | |
|---|---|---|---|---|---|---|---|---|---|
| | | Building | Contents | Building | Contents | Building | Contents | Building | Contents |
| 4 | 10 | 0.0% | 0.0% | 0.0% | 0.0% | 0.0% | 0.0% | 0.0% | 0.0% |
| | 15 | 0.0% | 0.0% | 0.0% | 0.0% | 0.0% | 0.0% | 0.0% | 0.0% |
| | 20 | 0.0% | 0.0% | 0.0% | 0.0% | 0.0% | 0.0% | 0.0% | 0.0% |
| 5 | 10 | 1.7% | 1.0% | 1.9% | 1.0% | 1.0% | 1.0% | 1.0% | 1.0% |
| | 15 | 0.7% | 0.0% | 0.9% | 0.0% | 0.0% | 0.0% | 0.0% | 0.0% |
| | 20 | 0.0% | 0.0% | 0.0% | 0.0% | 0.0% | 0.0% | 0.0% | 0.0% |
| 6 | 10 | 6.2% | 4.0% | 6.7% | 4.9% | 4.0% | 4.0% | 5.0% | 3.9% |
| | 15 | 3.5% | 2.0% | 3.8% | 2.1% | 3.0% | 2.0% | 2.9% | 1.9% |
| | 20 | 1.7% | 1.0% | 1.9% | 1.0% | 1.0% | 1.0% | 1.9% | 1.0% |
| 7 | 10 | 17.0% | 13.7% | 18.4% | 14.4% | 13.0% | 12.0% | 17.0% | 13.6% |
| | 15 | 11.2% | 8.7% | 12.1% | 9.0% | 9.0% | 8.0% | 10.6% | 8.7% |
| | 20 | 7.5% | 5.7% | 7.7% | 5.9% | 6.0% | 5.0% | 6.8% | 5.8% |



**Table 6: Estimated mean loss ratio of building based on structure type.**

| M | Focal Depth (Km) | Masonry | | Concrete | | Steel | | Wood | | Others | |
|---|---|---|---|---|---|---|---|---|---|---|---|
| | | Building | Contents | Building | Contents | Building | Contents | Building | Contents | Building | Contents |
| 4 | 10 | 0.0% | 0.0% | 0.0% | 0.0% | 0.0% | 0.0% | 0.0% | 0.0% | 0.0% | 0.0% |
| | 15 | 0.0% | 0.0% | 0.0% | 0.0% | 0.0% | 0.0% | 0.0% | 0.0% | 0.0% | 0.0% |
| | 20 | 0.0% | 0.0% | 0.0% | 0.0% | 0.0% | 0.0% | 0.0% | 0.0% | 0.0% | 0.0% |
| 5 | 10 | 2.0% | 1.0% | 1.0% | 1.0% | 1.0% | 1.0% | 1.0% | 1.0% | 1.0% | 0.0% |
| | 15 | 1.0% | 0.0% | 0.0% | 0.0% | 0.0% | 0.0% | 0.0% | 0.0% | 0.0% | 0.0% |
| | 20 | 0.1% | 0.0% | 0.0% | 0.0% | 0.0% | 0.0% | 0.0% | 0.0% | 0.0% | 0.0% |
| 6 | 10 | 7.0% | 4.0% | 6.0% | 5.0% | 5.0% | 5.2% | 4.0% | 4.0% | 5.0% | 2.0% |
| | 15 | 4.0% | 2.0% | 3.0% | 2.0% | 2.1% | 2.2% | 3.0% | 2.0% | 3.0% | 1.0% |
| | 20 | 3.0% | 1.0% | 1.0% | 1.0% | 1.0% | 1.0% | 2.0% | 1.0% | 2.0% | 0.0% |
| 7 | 10 | 18.0% | 13.0% | 19.0% | 15.0% | 16.9% | 14.8% | 13.0% | 14.0% | 16.0% | 8.0% |
| | 15 | 12.0% | 8.0% | 12.0% | 10.0% | 10.1% | 9.2% | 9.0% | 9.0% | 10.0% | 5.0% |
| | 20 | 8.1% | 5.0% | 7.0% | 6.0% | 6.1% | 6.0% | 6.0% | 6.0% | 7.0% | 3.0% |



**Table 7: Aggregated loss amount due to each scenario.**

| M | Focal Depth (Km) | Aggregated loss amount (Million USD) | | |
|---|---|---|---|---|
| | | Building | Contents | Total |
| 4 | 10 | 1,789 | 384 | 2,173 |
| | 15 | 583 | 110 | 694 |
| | 20 | 58 | 0 | 58 |
| 5 | 10 | 8,879 | 2,430 | 11,309 |
| | 15 | 4,330 | 1,089 | 5,419 |
| | 20 | 2,243 | 500 | 2,744 |
| 6 | 10 | 32,955 | 9,558 | 42,512 |
| | 15 | 17,974 | 5,234 | 23,208 |
| | 20 | 10,749 | 3,091 | 13,840 |
| 7 | 10 | 98,927 | 27,668 | 126,594 |
| | 15 | 61,120 | 17,861 | 78,980 |
| | 20 | 39,416 | 11,773 | 51,189 |





**FIGURE**

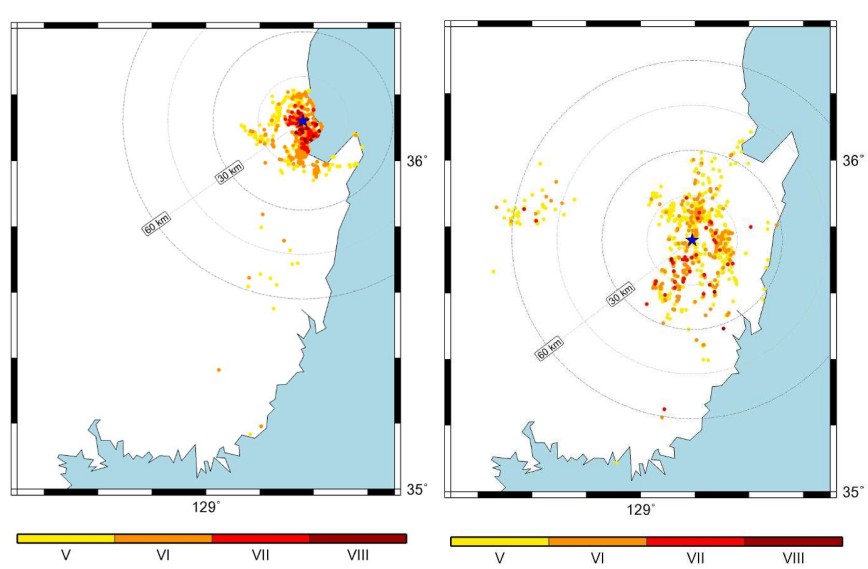

**Figure 1: MMI map due to Pohang earthquake (left) and Gyeongju earthquake (right). Source: KMA (2018).**





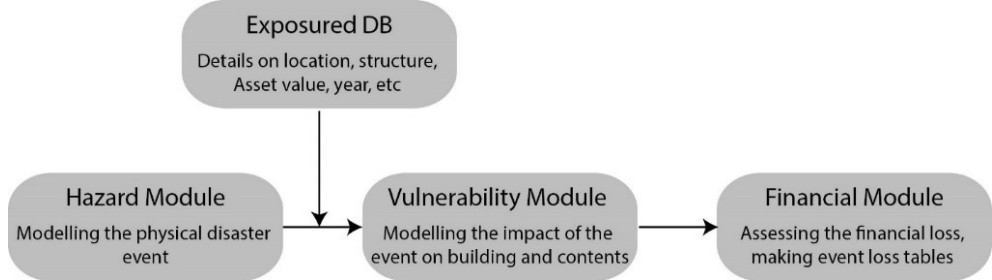

Figure 2: Procedure of a catastrophe model. Source: modified from Parodi (2014).



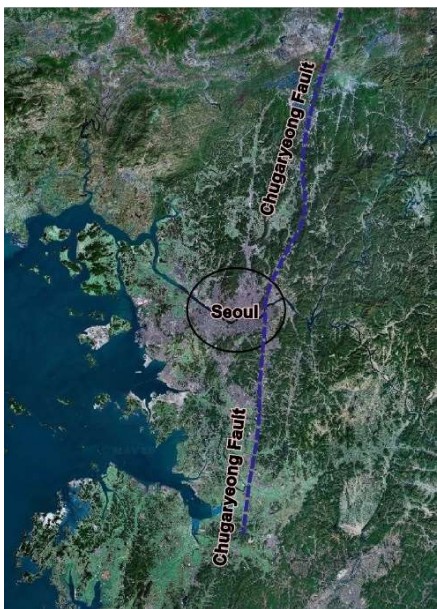

**Figure 3: Chugaryeong fault zone in the middle of the Korean peninsula. Source: Modified from Chung et al. (2014)**





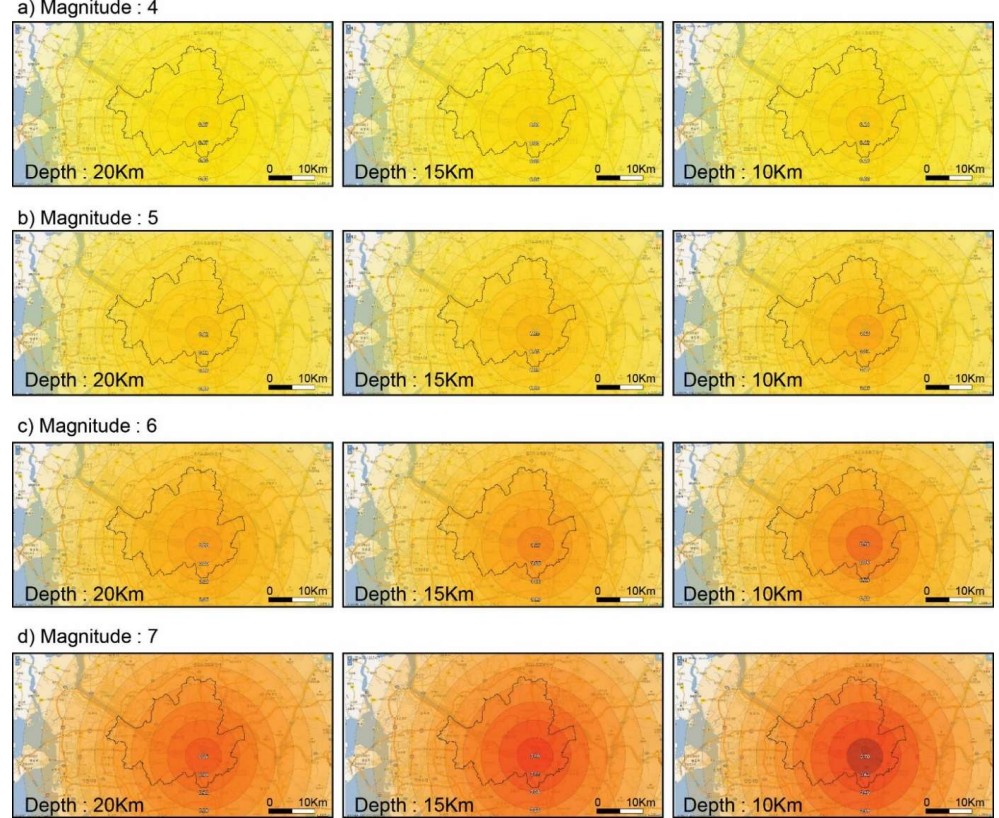

Figure 4: PGA hazard map of Seoul according to each scenario event.





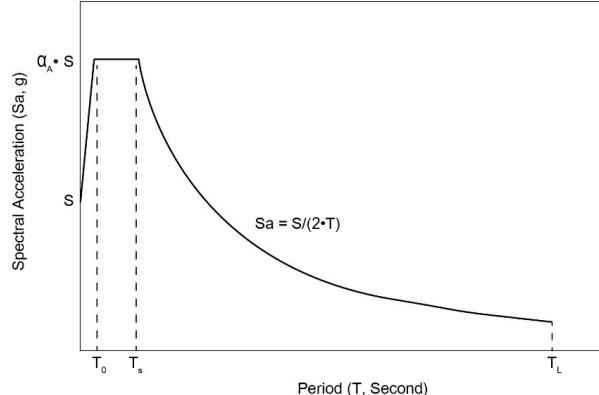

**Figure 5: Shape of standard response spectrum.**





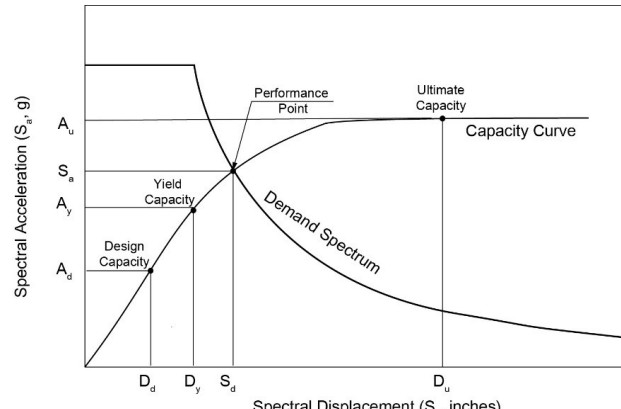

**Figure 6: Performance point according to intersection of capacity curve and demand spectrum.**




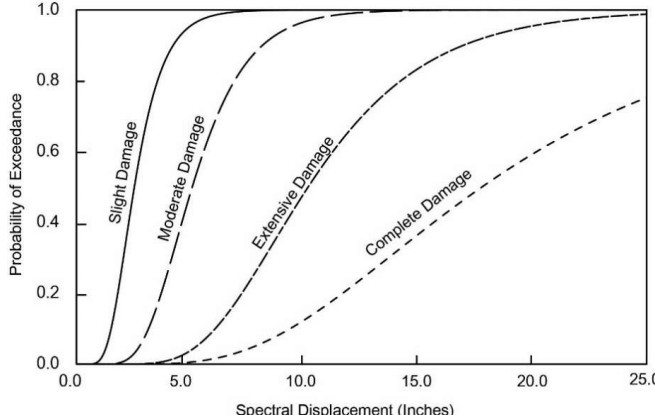

**Figure 7: Example of fragility curve for structural component.**



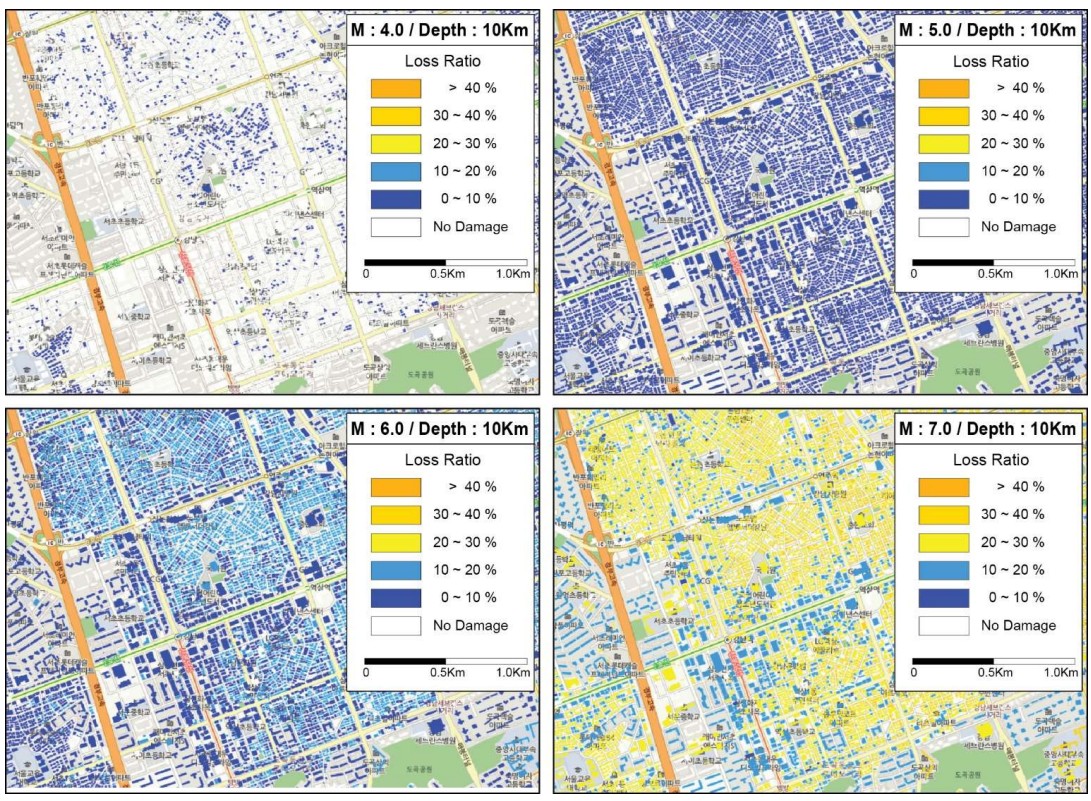

**Figure 8: Loss ratio map for each building by the scenario.**

