# Peer review of "Loss assessment of building and content damages from potential earthquake risk in Seoul, Korea"

_Natural Hazards and Earth System Sciences, 2018_

## Referee Comment (RC1) · Anonymous Referee #1 · 13 Jan 2019

This study evaluated risk based on probabilities of earthquake damages and their loss amount for Seoul city of Korea. The challenge of this study is an implementation of real and detailed data based on many earthquake scenarios considering active fault information. The contents in this study is worth to be published in the journal after major revision. Especially, more step-by-step details should be provided to efficiently understand loss amount evaluation clearly. The reviewer also addresses the following suggestions.

In Abstract, there are no general findings from this study. In Introduction, challenge of this study should be added. In Introduction, addition of literature review is required

to emphasize the challenge of this study. Numerate past research outcomes and their limitations. The results from this study is much different from those from the previous reports written in the Introduction. Please compare their results and explain the reasons of the difference. (Page 1 Line 31 and there after) Indicate specific magnitude scale in the manuscript with brief explanation of physics. (example: Richter local magnidue...) (Page 2 Line 23) Do not use ambigious expression in referencing figures and tables (replace "As shown in the above figure" with "As shown in Fig. 2") (Page 2 Line 30) Replace "36 the structure types" with "36 structure types"; Classification of structure types, occupancies, and seismic codes are not clearly explaned. (Table 1) In addtion to Distribution, add frequncies corresponding to eath distribution. (Table 2) There are three formulas in Table 2. It is necessary to either provide the other two formulas or erase contents of formulas I and II. (Page 3 Line 21) Indicate the reference time of building statistics of 6.3 million. (Page 3 Line 30) Replace "each building" with "each buliding damaged by earthquake." Reference is required for information provided by Korea Appraisal Board. (Page 4 Line 7) "Korea has been" should be "Korea has been considred as." Replace "neighboring countries" with "neighboring countries, such as Japan, China, and Taiwan." (Page 4 Line 8) Consider including the Yangsan Fault in your analysis. The fault might not have an influence on your research outcome, but it is worth to check its impact. (Page 4 Line 21) Abbreviation should be spelled out at its first appearance. "PGA" should be "Peak Ground Acceleration (PGA)." [Eq. (1)] To use minimum number of symbols, replace symbol S with PGA in Eq. (1). (Page 4 Line 27) Replace "the values of the coefficients" with "the values of the coefficients of C0, C1, C2, and C3." (Page 5 Line 22) "S is Peak Ground Acceleration (PGA) and $\alpha A$ is the amplification factor at the short period." should be indicated in Fig. 5. (Page 5 Line 26) "as shown table 3" should be "as shown in Table 3" (Page 6 Line 11) "zones: Zone I which includes Seoul area and Zone II" (Page 6 Line 13) For "zone 2A of UBC and the code of buildings built after 2009 is similar to zone 2B of UBC," indicate definition of Zones 2A and 2B. (Page 6 Line 29) Indicate definition of period T in Eq. (2). (Page 7 Line 4) "four 5 states;" should be "four 5 states:" (Page 7 Line 8) Subscription

is required for "Sd" and symbols in Eq. (3) (Page 7 Line 13) "is median value" should be "is the median value" (Page 7 Line 31) "Table 5 and 6" should be "Tables 5 and 6" (Page 8 Line 3) "isn't" should be "is not" (Page 8 Line 4) "higher" should be "larger" (Page 8 Line 5) "like" should be "such as" (Page 8 Line 6) Unified expression of dash use: "low-rise" and "low rise" are mixed. [Eq. (4)] There are mis fonts in subsription in Symbols in Eq. (4). (Page 9 Line 7) "% 15" should be "15%" (Tables 5, 6, and 7) "Km" should be "km"

---

## Short Comment (SC1) · 21 Jan 2019

1. In Abstract, there are no general findings from this study.

We added the following to abstract.

"The results of this study show that loss amounts due to potential earthquakes are significantly less than previous study. This is why the earthquake response spectrum and asset value of each building are applied as actual data from Korea."

2. In Introduction, challenge of this study should be added. In Introduction, addition of literature review is required to emphasize the challenge of this study. Numerate past

research outcomes and their limitations.

We added the following to introduction.

"This study differs from the previous studies in that it applies the actual building and insurance data and the observed seismic characteristics in Korea."

3. The results from this study is much different from those from the previous reports written in the Introduction. Please compare their results and explain the reasons of the difference.

We added the following to the result part.

"Nonetheless, the loss from the Mw 7.0 earthquake is only 4% compared to result of MPSS (2015) and the main reasons are as follows; 1) the duration of strong motion is applied as 0.6 second in the standard response spectrum in previous study, however 0.2 second reflected the recent earthquake characteristics in the Korean peninsula is applied in this study. 2) The replacement cost of each building is applied statistically to the actual insured data, but previous study was applied the replacement cost published in Square Foot Costs (RS Means, 2002) in USA. 3) This study don't consider indirect loss such as relocation expenses, income loss, rental income loss et al."

4. (Page 1 Line 31 and thereafter) Indicate specific magnitude scale in the manuscript with brief explanation of physics. (example: Richter local magnitude...)

We added the following to the text.

"The Richter magnitude scale (ML) is a unit based on logarithms calculated from the largest amplitude observed in the seismometer but it is difficult to measure accurately. In this study, the moment magnitude scale (Mw) is used, which it used by the United States Geological Survey (USGS) to calculate and report magnitudes for all modern large earthquakes is used."

5. (Page 2 Line 23) Do not use ambiguous expression in referencing figures and tables

(replace "As shown in the above figure" with "As shown in Fig. 2")

The problematic expression was corrected. (As shown in in Fig. 2)

6. (Page2 Line 30) Replace "36 the structure types" with "36 structure types"; Classification of structure types, occupancies, and seismic codes are not clearly explained.

We added the following to the text.

"The extracted data are classified into 36 structure types and 33 occupancies as same as the building type of HAZUS-MH, and divided into 3 seismic codes estimated based on comprehensive consideration of the construction year, total building area and occupancy."

7. (Table 1) In addition to Distribution, add frequencies corresponding to each distribution.

Table 1 was revised. In the text, the number of buildings is described as 6.3 million, This is a miswrite of about 630,000. This part was also revised.

8. (Table 2) There are three formulas in Table 2. It is necessary to either provide the other two formulas or erase contents of formulas I and II.

We revised it in consideration of the points indicated.

9. (Page 3 Line 21) Indicate the reference time of building statistics of 6.3 million.

We added a reference time in the text. (Seoul city as of 2016 database of building registration records.)

10. (Page 3 Line 30) Replace "each building" with "each building damaged by earthquake." Reference is required for information provided by Korea Appraisal Board.

We added a reference (Korea Appraisal Board (KAB): Construction Cost Table, 2016. (in Korean))

11. (Page 4 Line 7) "Korea has been" should be "Korea has been considered as."

Replace "neighboring countries" with "neighboring countries, such as Japan, China, and Taiwan." We revised it in consideration of the points indicated.

12. Abbreviation should be spelled out at its first appearance. "PGA" should be "Peak Ground Acceleration (PGA). We revised it in consideration of the points indicated.

13. [Eq. (1)] To use minimum number of symbols, replace symbol S with PGA in Eq. (1). We revised it in consideration of the points indicated.

14. (Page 4 Line 27) Replace "the values of the coefficients" with "the values of the coefficients of C0, C1, C2, and C3." We revised it in consideration of the points indicated.

15. (Page 5 Line 26) "as shown table 3" should be "as shown in Table 3" We revised it in consideration of the points indicated.

16. (Page 6 Line 11) "zones: Zone I which includes Seoul area and Zone II" We revised it in consideration of the points indicated.

17. (Page 7 Line 4) "four 5 states;" should be "four 5 states:" We revised it in consideration of the points indicated.

18. (Page 7 Line 8) Subscription is required for "Sd" and symbols in Eq. (3) We revised it in consideration of the points indicated.

19. (Page 7 Line 13) "is median value" should be "is the median value" We revised it in consideration of the points indicated.

20. (Page 7 Line 31) "Table 5 and 6" should be "Tables 5 and 6" We revised it in consideration of the points indicated.

21. (Page 8 Line 3) "isn't" should be "is not" We revised it in consideration of the points indicated.

22. (Page 8 Line 4) "higher" should be "larger" We revised it in consideration of the points indicated.

[Figure]

23. (Page 8 Line 5) "like" should be "such as" We revised it in consideration of the points indicated.

24. (Page 8 Line 6) Unified expression of dash use: "low-rise" and "low rise" are mixed. We revised it in consideration of the points indicated.

25. [Eq. (4)] There are mis fonts in subscription in Symbols in Eq. (4). We revised it in consideration of the points indicated.

26. (Page 9 Line 7) "% 15" should be "15%" We revised it in consideration of the points indicated.

27. (Tables 5, 6, and 7) "Km" should be "km" We revised it in consideration of the points indicated.

Thank you for your careful attention

Please also note the supplement to this comment:
https://www.nat-hazards-earth-syst-sci-discuss.net/nhess-2018-281/nhess-2018-281-SC1-supplement.pdf

**Supplement:**

**Loss assessment of building and content damages from potential earthquake risk in Seoul, Korea**

Wooil Choi[1], Jae-Woo Park[2] and Jinhwan Kim[2,3]

[1]Research Institute, LIG System, 442, Bongeunsa-ro, Gangnam-gu, Seoul, 06153, Korea
[2]Department of Civil and Environmental Engineering, Hanyang University, 222 Wangsimni-ro, Seongdong-gu, Seoul, 04763, Korea
[3]Multi Disaster Countermeasures Organization, Korea Institute of Civil Engineering and Building Technology, 283, Goyang-daero, Ilsanseo-gu, Goyang-si, Gyeonggi-do, 10223, Korea

*Correspondence to*: Jinhwan Kim (goethite@kict.re.kr)

**Abstract.** After the 2016 Gyeongju earthquake and the 2017 Pohang earthquake struck the Korean peninsula, securing financial stability for earthquake risk has become an important issue in Korea. Many domestic researchers are currently studying potential earthquake risk. However, empirical analysis and statistical approach are ambiguous in the case of Korea because no major earthquake has ever occurred on the Korean peninsula since Korean Meteorological Agency started monitoring earthquakes in 1978. This study focuses on evaluating possible losses due to earthquake risk in Seoul, the capital of Korea, by using catastrophe model methodology integrated with GIS (Geographic Information System). The building information such as structure and location is taken from the building registration database and the replacement cost for building is obtained from insurance information. As the seismic design code in KBC (Korea Building Code) is similar to the seismic design code of UBC (Uniform Building Code), the damage functions provided by HAZUS-MH are used to assess the damage state of each building in event of an earthquake. 12 earthquake scenarios are evaluated considering the distribution and characteristics of active fault zones in the Korean peninsula, and damages with loss amounts are calculated for each of the scenarios. The results of this study show that loss amounts due to potential earthquakes are significantly less than previous study. This is why the earthquake response spectrum and asset value of each building are applied as actual data from 
[revised manuscript text omitted]

---

## Referee Comment (RC2) · Anonymous Referee #2 · 20 Feb 2019

My opinion is that this paper differs from the previous paper in that it estimates the loss amount by applying the vulnerability function based on actual data of Korean buildings.

In addition, the damage estimation of this paper could be more realistic than the predicted from previous study considering the damage scale of Gyeongju and Pohang earthquakes.

However, there still needs some modification or additional explanations for better understaind like below.

1) The amount of damage predicted in the first and second revised manuscript is very different. Authors have to explain why the loss amount predicted in original manuscript

[Figure]

was revised.

2) The description of the methodology used isn't clear enough. It needs to be explained more clearly.

3) A few words used in this paper seem to be misleading in the interpretation, so it is necessary to replace them with words that cannot be interpreted in error.
* * *

---

## Author Comment (AC1) · 27 Feb 2019

1. The amount of damage predicted in the first and second revised manuscript is very different. Authors have to explain why the loss amount predicted in original manuscript was revised.

In this study, we estimated loss amounts based on the KRW. An error occurred converting the estimated loss amount to USD. That is why the loss amount is been revised.

2. The description of the methodology used isn't clear enough. It needs to be explained more clearly.

The methodology has been revised as follows to understand clearly.

(STEP 1) Information database of property which may be exposed to disaster should be constructed. (STEP 2) Hazard module for generation of a physical hazard map from a simulated event of disaster should be developed. (STEP 3) Vulnerability module to assess damage state of individual properties should be prepared by combining the information of exposed property and hazard intensity. (STEP 4) The financial module is implemented to quantify the damage of individual buildings into a monetary loss to predict a total loss amount.

3. A few words used in this paper seem to be misleading in the interpretation, so it is necessary to replace them with words that cannot be interpreted in error.

Some misleading words are been revised as follows.

(Page 3, Line 23) Replace "etc. affecting" with "and other minor considerations. Influencing"

(Page 4 Line 4) Replace "900 billion dollars" with "900 billion US dollars"

(Page 4 Line 5) Replace "between USD 100,000 to 1,000,000" with "0.1~1 million US dollars."

Thank you

Please also note the supplement to this comment:
https://www.nat-hazards-earth-syst-sci-discuss.net/nhess-2018-281/nhess-2018-281-AC1-supplement.pdf

**Supplement:**

**Loss assessment of building and content damages from potential earthquake risk in Seoul, Korea**

Wooil Choi[1], Jae-Woo Park[2] and Jinhwan Kim[2,3]

[1]Research Institute, LIG System, 442, Bongeunsa-ro, Gangnam-gu, Seoul, 06153, Korea
5  [2]Department of Civil and Environmental Engineering, Hanyang University, 222 Wangsimni-ro, Seongdong-gu, Seoul, 04763, Korea
[3]Multi Disaster Countermeasures Organization, Korea Institute of Civil Engineering and Building Technology, 283, Goyang-daero, Ilsanseo-gu, Goyang-si, Gyeonggi-do, 10223, Korea

10  *Correspondence to*: Jinhwan Kim (goethite@kict.re.kr)

**Abstract.** After the 2016 Gyeongju earthquake and the 2017 Pohang earthquake struck the Korean peninsula, securing financial stability for earthquake risk has become an important issue in Korea. Many domestic researchers are currently studying potential earthquake risk. However, empirical analysis and statistical approach are ambiguous in the case of Korea because no major earthquake has ever occurred on the Korean peninsula since Korean Meteorological Agency started
15  monitoring earthquakes in 1978. This study focuses on evaluating possible losses due to earthquake risk in Seoul, the capital of Korea, by using catastrophe model methodology integrated with GIS (Geographic Information System). The building information such as structure and location is taken from the building registration database and the replacement cost for building is obtained from insurance information. As the seismic design code in KBC (Korea Building Code) is similar to the seismic design code of UBC (Uniform Building Code), the damage functions provided by HAZUS-MH are used to assess the damage
20  state of each building in event of an earthquake. 12 earthquake scenarios are evaluated considering the distribution and characteristics of active fault zones in the Korean peninsula, and damages with loss amounts are calculated for each of the scenarios. The results of this study show that loss amounts due to potential earthquakes are significantly less than previous study. This is why the earthquake response spectrum and asset value of each building are applied as actual data from 
[revised manuscript text omitted]

---

## Author Response (AR1)

**RESPONSES TO COMMENTS**

The authors thank the referees for their constructive and valuable comments. Point-by-point responses to the referee comments are as follows. Referee comments are in black, and the authors' responses are in blue. Note that line numbers in the authors' replies are associated with line numbers in the revised manuscript.

**Referee 1.**

1. In Abstract, there are no general findings from this study.

**Response:** As the referee's recommendation, we added the following sentences to the abstract (Page 1, Line 23~25):

"The results of this study show that loss amounts due to potential earthquakes are significantly less than those of the previous studies. The challenge of this study is to implement earthquake response spectrum and to reflect actual asset values of buildings of the metropolitan city of Seoul in Korea."

2. In Introduction, challenge of this study should be added. In Introduction, addition of literature review is required to emphasize the challenge of this study. Numerate past research outcomes and their limitations.

**Response:** As the referee's recommendation, we added the following sentences to the introduction (Page 2, Line 14~15):

"This study differs from the previous studies in that it implements the actual building and insurance data and the observed seismic characteristics of Korea."

3. The results from this study are much different from those from the previous reports written in the introduction. Please compare their results and explain the reasons of the difference.

**Response:** As the referee's recommendation, we added the following sentences to the result part (Page 10, Line 5~12):

"Nonetheless, the loss from the $M_w$ 7.0 scale earthquake is only 4% compared to loss resulted from MPSS (2015). The main reasons of the difference in loss are as follows: (1) The duration of strong motion is applied as 0.6 seconds in the standard response spectrum in the previous studies; however, in this study, the duration of 0.2 seconds is implemented to reflect the characteristics of the recent earthquake occurred in the Korean peninsula; (2) The replacement costs of buildings are reflected in the analysis using statistics of the actual insured data, but the previous studies were used the replacement costs published in Square Foot Costs (RSMeans, 2002) in USA; and (3) This study did not consider indirect loss such as relocation expenses, income loss, and rental income loss."

4. (Page 2 Line 2 and thereafter) Indicate specific magnitude scale in the manuscript with brief explanation of physics. (example: Richter local magnitude...)

**Response:** As the referee's recommendation, we added the following sentences in Page 5, Line 9~12:

"The Richter magnitude scale ($M_L$) is a unit based on logarithms calculated from the largest amplitude observed in the seismometer but it is difficult to measure the amplitude accurately. In this study, the moment magnitude scale ($M_w$) is used, which was suggested by the United States Geological Survey (USGS) to calculate and report magnitudes for large earthquakes."

5. (Page 2 Line 30) Do not use ambiguous expression in referencing figures and tables (replace "As shown in the above figure" with "As shown in Fig. 2")

**Response:** Modification are made as the referee's recommendation. (Page 2, Line 30)

6. (Page3 Line 6) Replace "36 the structure types" with "36 structure types"; Classification of structure types, occupancies, and seismic codes are not clearly explained.

**Response:** As the referee's comment, we added the following sentences in Page 3, Line 6~9:

"The extracted data are classified into 36 structure types and 33 occupancies as same as the building type of HAZUS-MH, and divided into 3 seismic codes estimated based on comprehensive consideration of the construction year, total building area and occupancy. The details of classifications of 36 structure types and 33 occupancies are showed in Tables 1 and 2, respectively."

7. (Table 3) In addition to Distribution, add frequencies corresponding to each distribution.

**Response:** As the referee's recommendation, we revised Table 3. And in the text (Page 2 Line 16 and Page 4 Line 7), there exists a typo; therefore, the number of buildings is revised to 6.3 million from 630,000 (the number in the previous manuscript).

8. (Table 2) There are three formulas in Table 2. It is necessary to either provide the other two formulas or erase contents of formulas I and II.

**Response:** Regarding the reviewer's comment, the following are added for more clear understanding (in Page 5, Line 21~25) and the Table 2 was removed.

"The attenuation equation (or formula) of Eq. (1) proposed by MPSS (2012) is used in this study. The attenuation formula of MPSS requires four coefficients (or fitting parameters). In this study, the four coefficients in Eq. (1) of $C_0 = 5.0244$, $C_1 = 0.5442$, $C_2 = -1.0020$, and $C_3 = 0$ are assumed in the analysis as the combination of the coefficients resulted in least error in prediction of maximum ground acceleration."

9. (Page 4 Line 7) Indicate the reference time of building statistics of 630,000

**Response:** As the referee's recommendation, we added the following sentences in Page 4, Line 7:
"Seoul city as of 2016 database of building registration records"

10. (Page 4 Line 16) Replace "each building" with "each building damaged by earthquake." Reference is required for information provided by Korea Appraisal Board.

**Response:** As the referee's recommendation, we revised the sentences in Page 4 Line 16, and added a reference in Page 4 Line 17. (Korea Appraisal Board (KAB): Construction Cost Table, 2016. (in Korean))

11. (Page 4 Line 27) "Korea has been" should be "Korea has been considered as." Replace "neighboring countries" with "neighboring countries, such as Japan, China, and Taiwan."

**Response:** As the referee's recommendation, we revised the sentences in Page 4 Line 27~28.

12. Abbreviation should be spelled out at its first appearance. "PGA" should be "Peak Ground Acceleration (PGA).

**Response:** As the referee's recommendation, we revised the sentences in Page 5 Line 15.

13. [Eq. (1)] To use minimum number of symbols, replace symbol S with PGA in Eq. (1).

**Response:** As the referee's recommendation, we revised the sentences in Page 5 Line 23.

14. (Page 6 Line 23) "as shown table 3" should be "as shown in Table 3"

**Response:** As the referee's recommendation, we revised the sentences in Page 6 Line 23.

15. (Page 7 Line 7) "zones: Zone I which includes Seoul area and Zone II"

**Response:** As the referee's recommendation, we revised the sentences in Page 7 Line 7.

16. (Page 7 Line 33) "four 5 states;" should be "four 5 states:"

**Response:** As the referee's recommendation, we revised the sentences in Page 7 Line 33.

17. (Page 8 Line 3) Subscription is required for "Sd" and symbols in Eq. (3)

**Response:** As the referee's recommendation, we revised the sentences in Page 8 Line 3.

18. (Page 8 Line 8) "is median value" should be "is the median value"

**Response:** As the referee's recommendation, we revised the sentences in Page 8 Line 8.

19. (Page 8 Line 26) "Table 5 and 6" should be "Tables 5 and 6"

**Response:** As the referee's recommendation, we revised the sentences in Page 8 Line 26.

20. (Page 8 Line 30) "isn't" should be "is not"

**Response:** As the referee's recommendation, we revised the sentences in Page 8 Line 30.

21. (Page 9 Line 1) "higher" should be "larger"

**Response:** As the referee's recommendation, we revised the sentences in Page 9 Line 1.

22. (Page 9 Line 2) "like" should be "such as"

**Response:** As the referee's recommendation, we revised the sentences in Page 9 Line 2.

23. (Page 9 Line 1) Unified expression of dash use: "low-rise" and "low rise" are mixed.

**Response:** As the referee's recommendation, we revised the sentences in Page 9 Line 1.

24. [Eq. (4)] There are mis fonts in subscription in Symbols in Eq. (4).

**Response:** As the referee's recommendation, we revised the Eq. (4) in Page 9 Line 27.

25. (Page 10 Line 4) "% 15" should be "15%"

**Response:** As the referee's recommendation, we revised the sentences in Page 10 Line 4.

26. (Tables 6, 7, and 8) "Km" should be "km"

**Response:** As the referee's recommendation, we revised the sentences in Tables 6, 7 and 8.

**Referee 2.**

1. The amount of damage predicted in the first and second revised manuscript is very different. Authors have to explain why the loss amount predicted in original manuscript was revised.
**Response:** In this study, we estimated loss amounts based on the KRW. An error occurred converting the estimated loss amount to USD. That is why the loss amount has been revised.

2. The description of the methodology used isn't clear enough. It needs to be explained more clearly.
**Response:** As the referee's recommendation, we revised the sentences to understand clearly in Page 2 Line 30~31, Page 3 Line 16, Line 25~26, Line 31~32.

   "(STEP 1) Information database of property which may be exposed to disaster should be constructed.

(STEP 2) Hazard module for generation of a physical hazard map from a simulated event of disaster should be developed.

(STEP 3) Vulnerability module to assess damage state of individual properties should be prepared by combining the information of exposed property and hazard intensity.

(STEP 4) The financial module is implemented to quantify the damage of individual buildings into a monetary loss to predict a total loss amount."

3. A few words used in this paper seem to be misleading in the interpretation, so it is necessary to replace them with words that cannot be interpreted in error.

**Response:** As the referee's recommendation, we revised some misleading words (Page 4, Line 6)

Replace "etc. affecting" with "and other minor considerations. Influencing"

(Page 4 Line 18)

Replace "900 billion dollars" with "900 billion US dollars"

(Page 4 Line 19)

Replace "between USD 100,000 to 1,000,000" with "0.1~1 million US dollars."

Thank you for your valuable and constructive comments. The authors appreciate it.

[revised manuscript text omitted]